# Vitamin D and the Risks of Depression and Anxiety: An Observational Analysis and Genome-Wide Environment Interaction Study

**DOI:** 10.3390/nu13103343

**Published:** 2021-09-24

**Authors:** Zhen Zhang, Xuena Yang, Yumeng Jia, Yan Wen, Shiqiang Cheng, Peilin Meng, Chun’e Li, Huijie Zhang, Chuyu Pan, Jingxi Zhang, Yujing Chen, Feng Zhang

**Affiliations:** Key Laboratory of Trace Elements and Endemic Diseases of National Health and Family Planning Commission, School of Public Health, Health Science Center, Xi’an Jiaotong University, Xi’an 710000, China; zhang785334865@yeah.net (Z.Z.); smile940323@stu.xjtu.edu.cn (X.Y.); jia.yu.meng@163.com (Y.J.); wenyan@mail.xjtu.edu.cn (Y.W.); chengsq0701@stu.xjtu.edu.cn (S.C.); mengpeilin@stu.xjtu.edu.cn (P.M.); lichune@stu.xjtu.edu.cn (C.L.); zhj2020@stu.xjtu.edu.cn (H.Z.); panchuyu_dsa@163.com (C.P.); 3120315071@stu.xjtu.edu.cn (J.Z.); c18003409402@163.com (Y.C.)

**Keywords:** vitamin D, depression, anxiety, genome-wide association study, polygenic risk score, genome-wide environment interaction study

## Abstract

Previous studies have suggested that vitamin D (VD) was associated with psychiatric diseases, but efforts to elucidate the functional relevance of VD with depression and anxiety from genetic perspective have been limited. Based on the UK Biobank cohort, we first calculated polygenic risk score (PRS) for VD from genome-wide association study (GWAS) data of VD. Linear and logistic regression analysis were conducted to evaluate the associations of VD traits with depression and anxiety traits, respectively. Then, using individual genotype and phenotype data from the UK Biobank, genome-wide environment interaction studies (GWEIS) were performed to identify the potential effects of gene × VD interactions on the risks of depression and anxiety traits. In the UK Biobank cohort, we observed significant associations of blood VD level with depression and anxiety traits, as well as significant associations of VD PRS and depression and anxiety traits. GWEIS identified multiple candidate loci, such as rs114086183 (*p* = 4.11 × 10^−8^, LRRTM4) for self-reported depression status and rs149760119 (*p* = 3.88 × 10^−8^, GNB5) for self-reported anxiety status. Our study results suggested that VD was negatively associated with depression and anxiety. GWEIS identified multiple candidate genes interacting with VD, providing novel clues for understanding the biological mechanism potential associations between VD and psychiatric disorders.

## 1. Introduction

Psychiatric disorders are a group of complex diseases, which are mainly characterized by varying degrees of obstacles in mental activities such as cognition, emotion, willpower, and behavior [1]. A meta-analysis showed that overall global prevalence of psychiatric disorders was increased with odds ratio of 1.179 (95% CI: 1.065–1.305) [2]. Additionally, another study has found that the prevalence of the common mental illnesses is continuously rising, particularly in low- and middle-income countries, with many people suffering from depression and anxiety disorders simultaneously [3]. Depression and anxiety disorders are the most common mental disorders in the general population [4]. According to the latest report of World Health Organization, it is estimated that there are 322 million people (4.4% of the world’s population) living with depression and more than 260 million people (3.6% of the global population) affected by anxiety disorders [3]. Furthermore, Wang et al. observed that depression and anxiety were significantly associated with higher cancer incidence, cancer-specific mortality and all-cause mortality [5]. Traditional treatment of depression and anxiety can only relieve rather than cure the condition, causing a tremendous economic burden on individual families and high suicide rate. Therefore, it is urgent to find new ways to treat and prevent depression and anxiety disorders.

Vitamin D (VD) is a member of the steroid hormone family. As a necessary vitamin to human body, it is not only cheap, but also widely available. Besides being ingested from daily diet and VD supplements, it can also be synthesized from 7-dehydrocholesterol in the skin through ultraviolet radiation-b (UVB) [6]. Previous studies on VD have found that genetic and environmental factors can affect the status and metabolism of VD. The data from twin and family studies have suggested that circulating VD concentrations are partly determined by genetics, with heritability ranging from 23% to 80% [7]. Furthermore, genetic studies of VD have found that genetic variations and alterations (e.g., deletions, amplifications, inversions) in genes involved in VD metabolism, catabolism, transport, or binding to its receptors may affect VD levels [7]. Moreover, a study of gene-environment interactions with VD has showed that specific genetic variants associated with VD metabolism may be correlated with prostate cancer risk in VD-deficient patients [8]. As for the biological function of VD, previous studies have demonstrated that VD plays an important role in bone health, reproduction and fertility, and immune function [9]. Growing evidence shows that VD also exerts a great influence on the development and adult brain, such as maintaining calcium balance and signaling, regulating neurotrophic factors, providing neuroprotection, modulating neurotransmission, and promoting synaptic plasticity [6]. In addition, a recent systematic review has shown that VD deficiency in adulthood may also be associated with adverse brain-related outcomes [6]. The discovery of the functions of VD in the brain and the continuous confirmation of the effects of VD on disease provides a new research direction for the study of psychiatric disorders in recent years.

Although a lot of research has been devoted to exploring the relationship between VD and depression and anxiety, it is still controversial whether VD was associated with the two mental disorders [10]. Some studies have observed an association between VD and depression symptoms [11,12,13], while other studies have not found this association [14,15]. Additionally, previous studies on the association between VD and anxiety symptoms observed inconsistent association [13,16]. The discrepant results may be not only related to potential confounding factors, but also to complex etiology of psychiatric disorders. Previous studies on the pathogenesis of psychiatric disorders have found that they are associated with a variety of factors. For example, numerous studies have proved that inflammation is related to the pathogenesis of psychiatric disorders, and that the increase in inflammation will affect the occurrence and development of some psychiatric disorders [17,18]. Other studies have found that both diet and the gut microbiome have a strong influence on emotional behavior and neural processes [19]. Dietary patterns and changes in the microbiome can affect symptoms of depression and anxiety disorder and increase the risk of both disease through the microbiome gut–brain axis (MGBA) [20]. Moreover, some studies have shown that psychiatric disorders are caused by the interplay between genetic susceptibility and environmental risk factors [6,21]. It has been reported that the heritability of major depression [22] and current anxiety symptoms [23] was estimated to be 38% and 31%, respectively. However, previous studies on exploring the associations between VD and depression and anxiety only focused on the influence of environmental or genetic factors, usually not considering the interactions between them, which may underestimate the effects of VD on the risks of depression and anxiety.

Polygenic risk score (PRS) is proposed by running a GWAS on a study sample, selecting SNPs according to the relevant phenotypes, and creating the sum of phenotype related alleles (usually weighted by the SNP-specific coefficients from the GWAS) that can be evaluated in other replication sample [24]. PRS analysis can not only explore the genetic associations between various complex diseases and traits, but also assess the influence of susceptible loci on disease risks [25]. For example, PRS-related studies conducted by Psychiatric Genomics Consortium have found significant associations between PRS of symptom scale score and the risk of schizophrenia [26] as well as the efficacy of antipsychotic medication [27]. Recently, Revez et al. [28] conducted a GWAS of 417,580 UK Biobank study participants and identified 143 genetic loci associated with VD. Using the PRS of VD traits as instrumental variables, we can further explore the associations between the PRS of VD traits and psychiatric disorders.

GWAS study has strong ability in identifying susceptibility genetic loci associated with psychiatric disorders [29]. However, previous studies have shown that most of the phenotypic variation in complex diseases and traits cannot be explained solely by genetic factors, because phenotypic variation can also occur through genetic environment (G × E) interactions, in which the genotypes of different individuals vary in response to environmental stimuli [30]. Therefore, we further adopted the genome-wide environment interaction study (GWEIS), which can not only assess the effects of G × E interactions, but also evaluate the effects of genetic interactions on a genome-wide scale, helping to identify new genetic risk variants and understand the potential biological mechanisms [31]. For example, Rivera et al. found 53 and 34 single nucleotide polymorphisms (SNP_S_) in additive interactions with smoking in Lofgren’s syndrome (LS) and non-LS, respectively, but the association did not persist when assessing the effect of smoking on sarcoidosis without genetic information [32].

Utilizing the individual blood VD levels and calculated VD PRS data in UK Biobank cohort, we conducted regression analysis to evaluate the associations between VD (including blood VD levels and calculated VD PRS data) and depression status and anxiety status in the UK Biobank. Then, based on the results of regression analysis, genome-wide environment interaction studies (GWEIS) were performed to clarify the potential effects of gene × Vitamin D interactions on depression and anxiety.

## 2. Materials and Methods

### 2.1. UK Biobank Cohort

The UK Biobank (UKB) is a large population-based prospective cohort study, with health-linked information, both regarding phenotype and genotype, on approximately 500,000 participants aged between 40 and 69 years from all over the United Kingdom in 2006 and 2010 [33]. All participants were asked to report a series of health status and demographic information through questionnaires and interviews, and approved to use their anonymous data for any health-related research. Informed consent was provided by UKB from all participants. This study was approved by UKB (Application 46478) and obtained participants’ health-related records.

### 2.2. UK Biobank Phenotypes of Vitamin D

A total of 376,803 UKB participants’ blood samples were collected for quantitative measurement of 25(OH)D levels via chemiluminescent immunoassay (CLIA), and 343,334 (91.12%) individuals had their vitamin D 25(OH)D levels (UK Biobank data field: 30,890) measured. The analysis was limited to the population of white British individuals (UK Biobank data field: 21,000).

### 2.3. UK Biobank Phenotypes of Depression and Anxiety

The phenotypes of depression and anxiety were defined according to the previous study [34]. The selection criteria of case group were defined based on self-reports (UK Biobank data fields: 20,002; 20,126; and 20,544). In order to classify participants as much as possible, patient health questionnaire-9 (PHQ-9) [35], general anxiety disorder-7 (GAD-7) [36], and composite international diagnostic interview short-form (CIDI-SF) [37] were used as strict inclusion and exclusion criterion. PHQ-9 and GAD-7 are score scales for depression and anxiety, used to screen and measure the severity of depression and anxiety, respectively. PHQ-9 is a classification scale focusing on nine depression symptoms and signs, with a total score (0–27) [38], and GAD-7 is a classification scale focusing on seven anxious symptoms and signs, with a total score (0–21) [39]. Detailed classification of depression and anxiety are presented in the Appendix A. The selection criteria of the control group were as follows: without symptoms of depression and anxiety defined by CIDI and self -reported, depression PHQ scores and anxiety GAD scores ≤ 5, and without core symptoms.

### 2.4. UK Biobank Genotyping, Imputation and Quality Control

A total of 488,377 participants of UKB cohort were genotyped by either the Affymetrix UK BiLEVE Axiom Array or the Affymetrix UKB Axiom arrays [40]. The imputation and the quality control of these genotype results were carried out based on UK10K project reference panel [41] and Haplotype Reference Consortium (HRC) [42] reference panel. Then, we removed the participants who reported inconsistencies between self-reported gender and genetic gender, without ethic consents and imputation data. Additionally, we excluded variants with the Hardy–Weinberg equilibrium test *p* > 1.0 × 10^−5^, a minor allele frequency (MAF) of < 0.01, and a genotype missing rate of > 0.05. Ultimately, we used KING software to exclude the genetically related individuals.

### 2.5. GWAS Data of Vitamin D

The latest large-scale GWAS summary statistics of VD were used here. Briefly, this GWAS dataset detected 18,864 independent SNPs that were statistically associated with VD [28]. The genotype data were quality-controlled and imputed against the HRC and UK10K by the UKB group. Then, a linear mixed model GWAS was implemented in fastGWA to identify the genetic loci associated with 25OHD concentrations. Additionally, a rank-based reverse normal transformation (RINT) was applied to the phenotype, age, and gender. The genotyping batch and the first 40 ancestry PCs were used as covariates in the mixed model. In order to determine the independent associations, a conditional and joint (COJO) analysis [43] was employed on the GWAS results to explain the correlation structure between SNPs in the 10-Mb window (COJO default parameters). Detailed information of the GWAS can be obtained in the published study [28].

### 2.6. PRS Analysis of Vitamin D

Using the VD associated SNPs from the GWAS (*p* < 5 × 10^−8^) [28], the PRS of VD of each individual was calculated as the sum of the risk allele they carried, weighted by the effect size of the risk allele [28]. The PRS of VD was computed by PLINK2.0 [44], according to the formula:$$\mathrm{PRS}=\sum_{i=1}^{n}\beta_{i}{\mathrm{SNP}}_{\mathrm{im}}$$

PRS denotes the PRS value of VD for UKB subject; n and i, respectively, denote the total number of sample size and genetic markers; $\;\beta_{i}$ is the effect parameter of risk allele of the significant SNP associated with VD, which was obtained from the GWAS of VD [28]; and SNP_im_ is the dosage (0, 1, 2) of the risk allele of the SNP associated with VD [28].

### 2.7. Statistical Analysis

In the UK Biobank cohort, we evaluated associations between vitamin D and depression and anxiety through regression analysis. Specifically, logistic regression analysis was employed to evaluate the associations of self-reported depression and anxiety status with blood VD, VD PRS before COJO adjustment, and VD PRS after COJO adjustment. Linear regression analysis was conducted to test the associations of the PHQ-9 score and the GAD-7 score with blood VD, VD PRS before COJO adjustment, and VD PRS after COJO adjustment. In this regression analysis, blood VD, VD PRS before COJO adjustment, and VD PRS after COJO adjustment were used as independent variables. Self-reported depression, self-reported anxiety, PHQ-9 score, and GAD-7 score were used as outcome variables. Sex, age, and 10 principle components (PCs) of population structure were used as covariates in the regression analysis. A *p* < 0.05 indicated an association. All analyses were conducted by R.

### 2.8. Genome-Wide Environment Interaction Studies (GWEIS)

The generalized linear regression model of PLINK2.0 [45] was used to estimate the gene × VD interaction effects on the risk of depression and anxiety, using age, gender, and the first 10 PCs as covariates. According to previous research [46], we used PLINK2.0 genetic additive (ADD) models and selected high-quality SNPs through a quality control filters: SNPs with a low call rates (<0.90), low minor allele frequencies (<0.01), or low Hardy–Weinberg equilibrium exact test *p*-values (<0.01) were excluded. *p* < 5.0 × 10^−8^ and *p* < 5.0 × 10^−7^ were defined as significant and suggestive interactions, respectively. GWEIS results were visualized with the circular Manhattan plots generated by the “CMplot” R script. (https://github.com/YinLiLin/R-CMplot) (accessed on 15 February 2021).

## 3. Results

### 3.1. General Population Characteristics

#### 3.1.1. Characteristics of UK Biobank Subjects with Blood Vitamin D Data

For depression traits, with self-reported depression status as the outcome variable, a total of 110,744 participants answered the depression-related questions, and 52,766 were classified into depression group. With the depression PHQ score as the outcome variable, a total of 109,543 participants completed the questionnaire. For anxiety traits, with self-reported anxiety status as the outcome variable, a total of 98,784 participants answered the anxiety-related questions, and 19,759 were classified into anxiety group. With the anxiety GAD score as the outcome variable, a total of 110,023 participants completed the questionnaire.

#### 3.1.2. Characteristics of UK Biobank Subjects with Vitamin D PRS Data

In the self-reported depression status, a total of 121,685 participants answered depression-related questions, of which 58,349 were included in depression group; in the depression PHQ scores, a total of 120,033 participants completed the questionnaire. In the self-reported anxiety status, a total of 108,309 participants answered anxiety-related questions, of which 21,807 were classified into anxiety group. In addition, in the anxiety GAD scores, a total of 120,590 participants completed the questionnaire. Detailed information is shown in Table 1.

### 3.2. Regression Analysis Result

#### 3.2.1. Associations between Blood Vitamin D and Depression, Anxiety Traits in UK Biobank Cohort

Significant associations of blood VD level with self-reported depression status (odds ratio (OR) = 0.89, *p* = 5.92 × 10^−77^) and self-reported anxiety status (OR = 0.92, *p* = 1.46 × 10^−22^) were observed. Associations were also observed between the blood VD level, the depression PHQ score (Beta = −0.062, standard error (SE) = 0.003, *p* = 5.95 × 10^−96^), and the anxiety GAD score (Beta = −0.030, SE = 0.00, *p* = 1.21 × 10^−21^).

#### 3.2.2. Associations between Vitamin D PRS and Depression, Anxiety Traits in UK Biobank Cohort

We observed significant associations of VD PRS before COJO adjustment with self-reported depression status (OR = 0.99, *p* = 3.82 × 10^−2^), depression PHQ score (Beta = −0.0060, SE = 0.003, *p* = 3.25 × 10^−2^), and anxiety GAD score (Beta = −0.010, SE = 0.00, *p* = 4.36 × 10^−2^). In addition, we also observed significant associations of VD PRS after COJO adjustment with a self-reported depression status (OR = 0.99, *p* = 1.84 × 10^−2^), a depression PHQ score (Beta = −0.0070, SE = 0.0030, *p* = 9.15 × 10^−3^), and an anxiety GAD score (Beta = −0.010, SE = 0.00, *p* = 1.02 × 10^−2^). Detailed information is shown in Table 2.

### 3.3. GWEIS Analysis Results

For self-reported depression status, GWEIS identified a significant gene × VD PRS interaction (*p* < 5.0 × 10^−8^) at the LRRTM4 gene (rs114086183, *p* = 4.11 × 10^−8^). For self-reported anxiety status, significant gene × VD PRS interaction was detected at the GNB5 gene (rs149760119, *p* = 3.88 × 10^−8^). For the depression PHQ score, two significant gene × blood VD interactions were identified at SLC11A2 and HIGD1C (rs117102029, *p* = 4.02 × 10^−8^). For the anxiety GAD score, we detected multiple significant gene × VD trait interactions, such as SMYD3 (rs142593645, *p* = 2.51 × 10^−8^), SEMA3E (rs76440131, *p* = 2.80 × 10^−10^), and VTI1A (rs17266687, *p* = 3.09 × 10^−8^). Among them, three genes (SEMA3E, DOCK8, TMCO3) were identified by VD PRS before and after COJO adjustment. The visualization of the results is shown in Figure 1, Figure 2 and Figure 3. Additional detailed results are shown in Table 3.

## 4. Discussion

Since many complex diseases are associated with thousands of genetic variations, GWAS study merely calculates the association between a single SNP and a phenotype, which could easily lead to a decline in the interpretation of phenotypes influenced by multiple genetic variations. In this study, using individual blood VD level and calculated VD PRS data, we systematically evaluated the associations of VD traits with depression and anxiety traits. Then, we conducted GWEIS to clarify the potential effects of gene × VD interaction on depression and anxiety traits. We observed significant associations between VD and depression and anxiety traits in the UK Biobank cohort, and GWEIS analysis identified the effects of multiple significant gene × VD interactions on depression and anxiety traits.

As mentioned before, previous studies on VD and psychiatric disorders merely focused on the effects of environmental or genetic factors on the risks of depression and anxiety, usually without considering the interaction between them. For anxiety, multiple studies have observed that VD level was associated with anxiety [47], and VD supplementation can significantly improve patients’ anxiety symptoms after adjusting for covariates known to affect VD level [48]. However, there is also controversy about the association between VD and depression. For instance, Zhao et al. found that VD was not associated with an increased risk of depression after adjusting for potential confounders, such as age, gender, race, physical activity, alcohol use, and chronic diseases [49]. Some studies conducted in different regions failed to observe the association between VD and depression when controlled for potential confounding factors [50,51]. In contrast, Milaneschi et al. observed that low VD levels were associated with the presence and severity of depression, suggesting that VD represented a potential biological vulnerability for depression [52]. Additionally, a prospective association study of VD and depression using UKB cohort data found that both vitamin D deficiency and insufficiency may be risk factors for new onset depression in middle-aged adults [53]. Our results support an association between VD traits and depression and anxiety traits in the UK Biobank cohort, particularly from a genetic perspective. It is important to note that our study found an association between VD and depression and anxiety; however, further research is needed to determine whether there was a causal association and in what direction.

Currently, to the best of our knowledge, there are limited researches to explore the genetic mechanism affecting the link between VD and depression and anxiety. Therefore, we performed GWEIS and identified multiple candidate genes interacting with VD, which are implicated in the brain or neural regulation and pathology, such as LRRTM4 for depression status and GNB5 for anxiety status. The LRRTM4 is a new four-membered family of genes from human and mice. Its main function is to encode a putative leucine-rich repeat transmembrane protein, which can not only facilitate the development of glutamate synapses, but also regulate many cellular events during nervous system development and disease [54,55]. In animal experiments, it has been found that LRRTM4 is expressed in many brain regions and nervous system neurons, suggesting that LRRTM4 plays a vital role in the development and maintenance of the vertebrate nervous system [54]. In addition, the role of VD in regulating brain axon growth has been observed in previous studies [56], and prenatal VD deficiency has been shown to alter many genes involved in synaptic plasticity [57]. Whether VD deficiency alters the LRRTM4 gene remains elusive and need further studies.

For anxiety status, the identified GNB5 gene is the G protein subunit beta 5 (Gβ5) that encodes a heterotrimeric GTP binding protein. Gβ5 is enriched in the central nervous system. Its main function is to form a complex with regulatory factors of the G protein signal transduction protein family, thereby regulating and affecting the neurotransmitter signal transduction of many neurobehavioral results [58]. A previous study found that VD can affect adult brain development and function through signal transduction [59]. Furthermore, a study of the expression of genes associated with Alzheimer’s disease in the presence of VD deficiency found that GNB5 expression was significantly reduced [60]. We may infer that VD can regulate G-protein-mediated signaling in the brain by influencing the GNB5 gene [60]. In addition, previous studies have indicated that GNB5 gene mutations can lead to severe speech disorders, motor delays, and attention deficit hyperactivity disorder (ADHD) as the main manifestations of recessive neurodevelopmental disorders [58,61].

For the anxiety GAD score, our GWEIS results showed that multiple genes have significant interactions with vitamin D. DPP6 is a single-channel type II transmembrane protein expressed in the brain, which mainly regulates the dendritic excitability of hippocampal neurons [62]. Cacace et al. found that DPP6 is a new genetic factor in dementia. DPP6 is involved in a variety of cellular pathways, including neurogenesis and neuronal excitability, and its deletion has been associated with low intelligence and neurodevelopmental disorders [63]. Another study in an animal model also found that DPP6 deletion affects hippocampal synaptic development and leads to behavioral impairments in recognition, learning, and memory [64]. Similar to the DPP6 gene, Tang’s study found that VTI1A is mainly involved in neuronal development and neurotransmission, and mutations are likely to lead to neurological dysfunction and neurological diseases [65]. In common genes identified by VD PRS before and after COJO adjustment, SEMA3E is a member of the signaling family that binds directly to the receptor Plexin-D1 to secrete brain signals. A study found that SEMA3E-Plexin-D1 signaling is not only involved in axon growth and guidance, but also determines synaptic recognition and specificity in multiple parts of the nervous system [66]. Although these genes have been found to play a certain role in the development and conduction of the nervous system, the potential biological mechanism of the interaction between these genes and VD to affect the nervous system function and disease has not been found, which needs further research and confirmation.

In addition, we identified multiple candidate genes which interacted with vitamin D for the depression PHQ score, such as SLC11A2 and HIGD1C. The SLC11A2 gene, also called DMT1, is an iron-responsive gene mainly involved in iron absorption [67]. Mutations in this gene will affect the changes in the body’s iron content. Bastian et al. [68] found that iron deficiency in the early life can damage the expression of hippocampal neuron genes, leading to long-term neurological dysfunction. Saadat et al. [69] observed that the TT genotype and the T allele of the 1254T > C polymorphism in the DMT1 gene may be a risk factor for Parkinson’s disease. At present, few researches were conducted to explain the function and role of the HIGD1C gene. However, the HIGD1A gene, which came from the same family as the HIGD1C gene (HIG1 hypoxia-inducible domain family), has been found to be related to the nervous system in previous studies. Research conducted by López’s et al. found that the HIGD1A gene is not only widely expressed in the rat brain, but also may play a protective role in certain areas of the central nervous system [70]. Nevertheless, no studies have shown a direct link between VD and the effects of SLC11A2 and HIGD1C genes on the nervous system and psychiatric disorders. It is worth mentioning that previous studies have found the important role of VD in the brain and nervous system, such as VD differentiates brain cells [71], which regulates axon growth [56] and can regulate calcium signaling [72]. VD can not only affect adult brain development and function through signal transduction, but also affect the nutritional support factors of developing and mature neurons and prevent the production of reactive oxygen species. These all support the importance of VD for development and function of human brain.

Molecular genetic studies have confirmed the presence of widespread pleiotropy across psychiatric disorders [73]. Previous studies have found that depression and anxiety are highly comorbid and share a common underlying basis, including symptom overlap, potential negative affectivity, shared familial risk, stress, negative cognitions, and similar neural-circuitry dysfunction related to emotion regulation [74,75,76]. According to Gray and McNaughton’s theory, this comorbidity is caused by the recursive interconnection of brain regions that connect fear, anxiety, and panic, as well as hereditary personality traits such as neuroticism [77]. Twin and familial studies have shown that comorbidity of depression and anxiety disorders is largely explained by shared genetic risk factors [77]. However, a recent factor analysis and genomic structural equation modelling study on depression and anxiety found that depressive and anxiety symptoms could be affected by different factors, although the genetic correlation between the factors was high [78]. In this study, we further compared and identified genetic loci between depression and anxiety; no overlapping loci were found, suggesting that VD may have different biological mechanisms in depression and anxiety. It is considered that environmental exposure can contribute to the development of depression and anxiety through different molecular mechanisms [79]. Furthermore, based on the results of genomic structural equation modeling [78] and the differences in etiology and pathogenesis of depression and anxiety [80], it is reasonable to infer that few genetic loci interacting with VD promote the occurrence and development of depression and anxiety at the same time. Genetic research that assesses the link between VD and psychiatric disorders are limited, and further exploration is needed to confirm our findings.

It is worth noting that our study has some limitations. First, all research data in our study were derived from the UK Biobank, and the research participants were limited to people of European descent. Due to different genetic backgrounds, the results of this study should be interpreted with caution when applying the results to other populations. Secondly, in our research, we mainly used self-reports and related questionnaire scores to characterize depression and anxiety states. Since there is no systematic method to classify all the symptoms, the self-reported analysis results may not be completely consistent with the analysis results of questionnaire scores; in addition, self-reported results may increase the possibility of measurement error and recall bias. Due to the lack of temporal sequence between variables and the absence of Mendelian Randomization study, it is not possible to draw evidence for causality directly, resulting in the lack of demonstration strength of the study. Finally, there is a lack of relevant researches to investigate the influence of the identified SNPs on the biological mechanisms of depression and anxiety. More large sample prospective studies and biological studies are needed to confirm our results and elucidate the potential role of new genetic variants in the pathogenesis of psychiatric disorders.

## 5. Conclusions

In summary, through regression analysis and GWEIS analysis, we observed that the VD was negatively associated with depression and anxiety, and further GWEIS analysis identified multiple candidate genes related to depression and anxiety. The interaction effects observed from the results provide new direction for understanding the genetic research of psychiatric disorders. Further research is needed to clarify the underlying mechanism of gene × VD interaction effects for psychiatric disorders.

## Figures and Tables

**Figure 1 nutrients-13-03343-f001:**
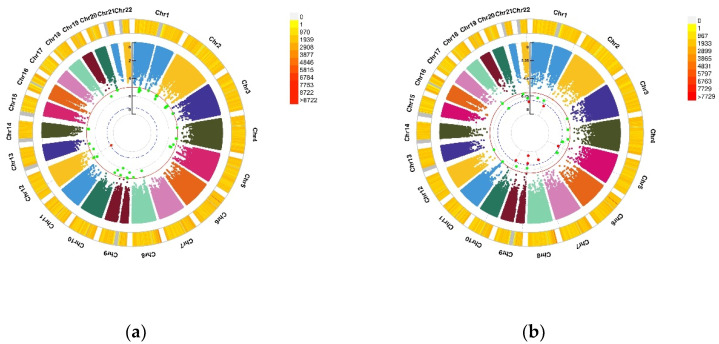
Genomic regions interacting with blood VD for the PHQ score and the GAD score. (**a**) Depression (PHQ score), A SNP allele was found to significantly interact with blood VD in depression (PHQ score); (**b**) Anxiety (GAD score), seven independent SNP alleles were found to significantly interact with blood VD in anxiety disorder (GAD score). From the center, the first circos depicts the −log10 *p*-values of each variant due to double exposure, i.e., the effect of both SNP allele and blood VD. The second circos shows chromosome density. Red dots represent the *p* < 5 × 10^−8^ and green dots represent *p* < 1 × 10^−7^. The figure was generated using the “CMplot” R script (https://github.com/YinLiLin/R-CMplot) (accessed on 15 February 2021).

**Figure 2 nutrients-13-03343-f002:**
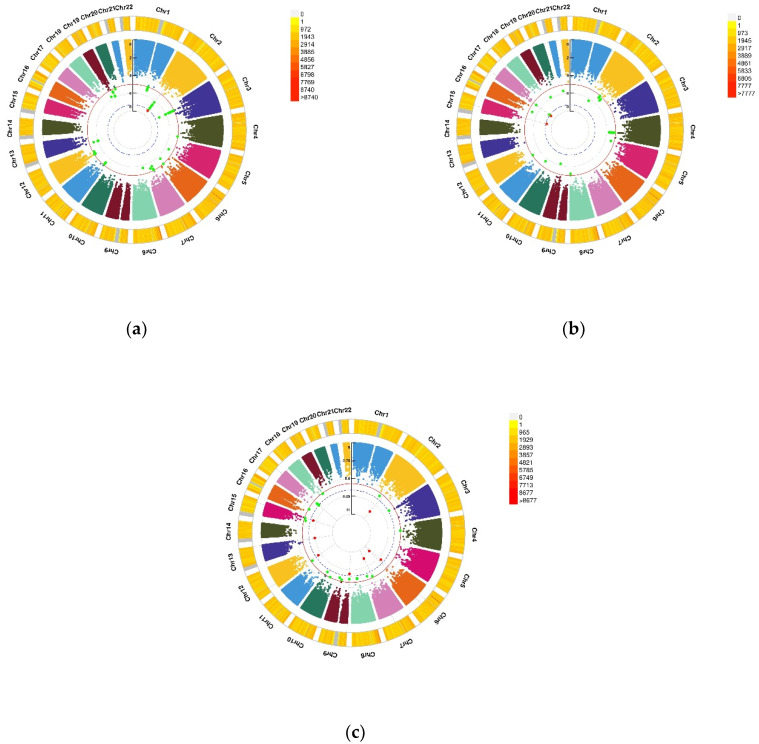
Genomic regions interacting with VD PRS after COJO adjustment for depression status, anxiety status, and GAD score. (**a**) Depression status, a SNP allele was found to significantly interact with blood VD in depression status; (**b**) Anxiety status, 2 independent SNP alleles were found to significantly interact with blood VD in anxiety status; (**c**) Anxiety (GAD score), 8 independent SNP alleles interacted significantly with blood VD in anxiety disorders (GAD score). From the center, the first circos depicts the −log10 *p*-values of each variant due to double exposure, i.e., the effect of both SNP allele and VD PRS after COJO adjustment. The second circos shows chromosome density. Red dots represent the *p* < 5 × 10^−8^ and green dots represent *p* < 1 × 10^−7^. The figure was generated using the “CMplot” R script (https://github.com/YinLiLin/R-CMplot) (accessed on 15 February 2021).

**Figure 3 nutrients-13-03343-f003:**
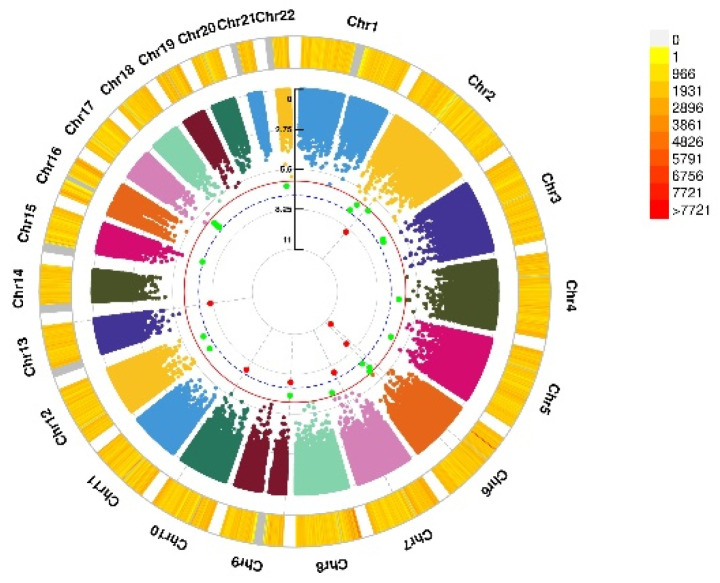
Genomic regions interacting with VD PRS before COJO adjustment for GAD score. This graph shows 7 independent SNP alleles interacting significantly with blood VD in anxiety disorders (GAD score). From the center, the first circos depicts the −log10 *p*-values of each variant due to double exposure, i.e., the effect of both SNP allele and VD PRS before COJO adjustment. The second circos shows chromosome density. Red dots represent the *p* < 5 × 10^−8^ and green dots represent *p* < 1 × 10^−7^. The figure was generated using the “CMplot” R script (https://github.com/YinLiLin/R-CMplot) (accessed on 15 February 2021).

**Table 1 nutrients-13-03343-t001:** General population characteristics of this study participants from UK Biobank.

Outcome Variable	Independent Variable	Number/ (Case/Control)	Sex (Female)	Age ± SD
Depression status	Blood VD	52,766/57,978	61,458 (55.50%)	56.40 ±7.68
	VDPRS After COJO	58,349/63,336	68,365 (56.18%)	56.47 ± 7.65
	VDPRS Before COJO	58,349/63,336	68,365 (56.18%)	56.47 ± 7.65
Anxiety status	Blood VD	19,759/79,025	53,541 (54.20%)	56.42 ± 7.60
	VDPRS After COJO	21,807/86,502	59,453 (54.89%)	56.50 ± 7.57
	VDPRS Before COJO	21,807/86,502	59,453 (54.89%)	56.50 ± 7.57
Depression (PHQ score)	Blood VD	109,543	60,377 (55.12%)	56.16 ± 7.65
	VDPRS After COJO	120,033	66,934 (55.76%)	56.24 ± 7.62
	VDPRS Before COJO	120,033	66,934 (55.76%)	56.24 ± 7.62
Anxiety (GAD score)	Blood VD	110,023	60,629 (55.11%)	56.15 ± 7.65
	VDPRS After COJO VDPRS Before COJO	120,590 120,590	67,235 (55.76%) 67,235 (55.76%)	56.23 ± 7.61 56.23 ± 7.61

Abbreviations: VD, Vitamin D; VDPRS, polygenic risk score of vitamin D; COJO, conditional and joint analysis; SD, age was described as mean ± standard deviation (SD); PHQ score, patient health questionnaire (PHQ) is used to describe the depression; GAD score, general anxiety disorder (GAD) is used to describe the anxiety of the participants.

**Table 2 nutrients-13-03343-t002:** The associations between Vitamin D traits and traits of depression and anxiety.

Outcome Variable	Independent Variable	Beta	SE	T	*p*−Value	OR
Depression status	Blood VD	−0.12	0.01	−18.57	5.92 × 10^−77^	0.89
	VDPRS After COJO	−0.014	0.006	−2.36	1.84 × 10^−2^	0.99
	VDPRS Before COJO	−0.012	0.006	−2.07	3.82 × 10^−2^	0.99
Anxiety status	Blood VD	−0.080	0.01	−9.77	1.46 × 10^−22^	0.92
	VDPRS After COJO	0.00	0.01	−0.29	7.71 × 10^−1^	1.00
	VDPRS Before COJO	0.00	0.01	−0.32	7.47 × 10^−1^	1.00
Depression (PHQ score)	Blood VD	−0.062	0.003	−20.81	5.95 × 10^−96^	–
	VDPRS After COJO	−0.007	0.003	−2.61	9.15 × 10^−3^	–
	VDPRS Before COJO	−0.006	0.003	−2.14	3.25 × 10^−2^	–
Anxiety (GAD score)	Blood VD	−0.030	0.00	−9.56	1.21 × 10^−21^	–
	VDPRS After COJO VDPRS	−0.010	0.00	−2.57	1.02 × 10^−2^	–
	Before COJO	−0.010	0.00	−2.02	4.36 × 10^−2^	–

Abbreviations: SE, standard error; T, *t*−test; OR, odd ratios; VD, vitamin D; VDPRS, polygenic risk score of vitamin D; COJO, conditional and joint analysis; PHQ score, patient health questionnaire (PHQ) is used to describe the depression; GAD score, general anxiety disorder (GAD) is used to describe the anxiety of the participants. Note. Logistic regression was used to evaluate the association between blood VD, VD PRS before COJO adjustment, VD PRS after COJO adjustment and self-reported depression and anxiety. Linear regression was used to evaluate the association between blood VD, VD PRS before COJO adjustment, VD PRS after COJO adjustment and PHQ score, GAD score.

**Table 3 nutrients-13-03343-t003:** Summary of gene−environment interaction analysis among SNP and VD traits for depression and anxiety traits.

	CHR	SNP	Model	Beta	SE	Gene	*p*−Value
Depression status	2	rs114086183	ADD ×VD PRS afterCOJO	0.16	0.029	LRRTM4	4.11 × 10^−8^
Anxiety status	15	rs149760119	ADD × VD PRS afterCOJO	−0.22	0.04	GNB5	3.88 × 10^−8^
Depression (PHQ score)	12	rs117102029	ADD × VD blood	0.01	0.002	SLC11A2, HIGD1C	4.02 × 10^−8^
Anxiety (GAD score)	1	rs142593645	ADD × VD blood	1.52	0.27	SMYD3	2.51 × 10^−8^
	7	rs13228257	ADD × VD blood	−1.25	0.22	DPP6	1.45 × 10^−8^
	7	rs76440131	ADD × VD PRS afterCOJO	−2.64	0.42	SEMA3E	2.80 × 10^−10^
	9	rs78029983	ADD × VD PRS afterCOJO	−1.08	0.19	DOCK8	2.43 × 10^−10^
	13	rs76004204	ADD × VD PRS afterCOJO	2.17	0.37	TMCO3	6.38 × 10^−9^
	7	rs76440131	ADD × VD PRS beforeCOJO	−2.16	0.38	SEMA3E	1.76 × 10^−8^
	9	rs78029983	ADD × VD PRS beforeCOJO	−1.10	0.20	DOCK8	2.10 × 10^−8^
	10	rs17266687	ADD × VD PRS beforeCOJO	−1.20	0.21	VTI1A	2.48 × 10^−8^
	13	rs76004204	ADD × VD PRS beforeCOJO	1.97	0.34	TMCO3	8.89 × 10^−9^

Abbreviations: CHR, chromosome; SNP, single nucleotide polymorphism; SE, standard error; ADD, additive effect; VD, vitamin D; VDPRS, polygenic risk score of vitamin D; COJO, conditional and joint analysis; PHQ score, patient health questionnaire (PHQ) is used to describe the depression; GAD score, general anxiety disorder (GAD) is used to describe the anxiety of the participants; *p*−value, estimates of the effect of interaction on depression and anxiety traits by using ADD × VD traits.

## Data Availability

The UK Biobank data are available through the UK Biobank Access Management System https://www.ukbiobank.ac.uk/ (accessed on 20 December 2020). We will return the derived data fields following UK Biobank policy; in due course, they will be available through the UK Biobank Access Management System.

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
