# Peer review of "Vitamin D and the Risks of Depression and Anxiety: An Observational Analysis and Genome-Wide Environment Interaction Study"

_nutrients, 2021, doi:10.3390/nu13103343_

Round 1
Reviewer 1 Report
Dear author, Congratulations on the manuscript. Given the relevance of the subject and after reading it, my suggestion for improvement is to improve one's position in relation to genetics, environment or other factors, because as a reader the position of the manuscript regarding vitamin D has not really been clear to me. His manuscript presents good methodology, good literature and conceptualization, but the discourse both in the introduction and in the discussion, since there is so much controversy on this topic (genetic or environmental) is not clear to me. Maybe the way to express things. That's my suggestion, and congratulations on this groundbreaking manuscript. All the best.
Author Response
请参阅附件。

Reviewer 2 Report
30 August 2021
Review on the manuscript titled “Vitamin D and the risks of depression and anxiety: an observational analysis and genome-wide environment interaction study” by Zhang Z et al., submitted to Nutrients.
Manuscript ID: nutrients-1374859
Dear Authors,
Vitamin D is linked to psychiatric diseases, but little is known about the genetic aspects of vitamin D in depression and anxiety. The authors studied the relationship of vitamin D with depression and anxiety and the interactions between gene and vitamin D by calculating polygenic risk score (PRS) from genome-wide association study and from genome-wide environment interaction studies, respectively. The authors revealed that vitamin D is negatively correlated with depression and anxiety, concluding the presence of genetic factors regarding vitamin D in psychiatric diseases.
Please reconsider the following parts:
- A graphical abstract is highly recommended.
- Pages 1-3, Abstract, Introduction: Please clarify if depression and anxiety as symptoms or psychiatric disorders. Please present a rationale to limit to and group the two in this study.
- Page 1, Keywords: Please list more keywords up to ten.
- Pages 2, Paragraph 2: The authors limit the pathogenesis of depression and anxiety to only environmental and genetic factors. There are many factors linking between them such as inflammation in the pathogenesis. The brief description is expected regarding the pathogenesis of depression and anxiety. Suggested refence: https://doi.org/10.3390/biomedicines9070734.
- Pages 5,6 Tables 1-3: Please define the abbreviations in the caption such as VD, VDPRS, and COJO.
- Page 7,8, Figures 1-3: Please briefly describe the results in each caption.
- Pages 4-8, Results: Please present more statistical data such as effect size measure, odds ratios, correlation coefficients relative risk, and coefficient of determination in regression analysis, among others. And numerical results should be reported in open form such as in the tables.
- Pages 8-11, Discussion, Conclusion: Please describe potentials in the present study, the ultimate goal, research or knowledge needed to achieve, and the biggest challenge in this goal, among others.
- Pages 8-11, Discussion: It deserves to discuss more on depression and anxiety as components and a spectrum of symptoms including cognitive symptoms. Suggested reference: https://doi.org/10.3390/biomedicines9050517; https://doi.org/10.3390/biomedicines9040340.
- Pages 8-11, Discussion: Please present a figure summarizing the discussion.
The manuscript contains three figures, three tables, and 66 references. The manuscript carries important value presenting the relationships and interactions of vitamin D and genes in depression and anxiety. I reconsider this manuscript for publication after major revision.
Round 2
Reviewer 2 Report
19 September 2021
Review on the manuscript titled “Vitamin D and the risks of depression and anxiety: an observational analysis and genome-wide environment interaction study” by Zhang Z et al., submitted to Nutrients.
Manuscript ID: nutrients-1374859
Dear Authors,
Vitamin D is linked to psychiatric diseases, but little is known about the genetic aspects of vitamin D in depression and anxiety. The authors studied the relationship of vitamin D with depression and anxiety and the interactions between gene and vitamin D by calculating polygenic risk score (PRS) from genome-wide association study and from genome-wide environment interaction studies, respectively. The authors revealed that vitamin D is negatively correlated with depression and anxiety, concluding the presence of genetic factors regarding vitamin D in psychiatric diseases.
The manuscript contains three figures, three tables, and 80 references. The Authors addressed their response properly and revised the manuscript accordingly. The quality of the manuscript is substantially improved. Please insert “The Supporting Information Figure 1.” Into the body of the manuscript. Or it can be the graphical abstract. The manuscript carries important value presenting the relationships and interactions of vitamin D and genes in depression and anxiety. I recommend this manuscript for publication after minor revision.
I declare no conflict of interest regarding this manuscript.
Best regards,
Author Response
Response to Reviewer 2 Comments
The manuscript contains three figures, three tables, and 80 references. The Authors addressed their response properly and revised the manuscript accordingly. The quality of the manuscript is substantially improved. Please insert “The Supporting Information Figure 1.” Into the body of the manuscript. Or it can be the graphical abstract. The manuscript carries important value presenting the relationships and interactions of vitamin D and genes in depression and anxiety. I recommend this manuscript for publication after minor revision.
Point 1: Please insert “The Supporting Information Figure 1.” Into the body of the manuscript. Or it can be the graphical abstract.
Response 1: Thank you for your helpful comments.
According to your suggestion, we inserted the summary figure of the discussion section, the original " The Supporting Information Figure 1", as the graphical abstract in our article.
Thanks!